# P-RAG: Progressive Retrieval Augmented Generation for Planning on Embodied Everyday Tasks

## ABSTRACT

Embodied Everyday Task is a popular task in the embodied AI community, requiring agents to make a sequence of actions based on natural language instructions and visual observations. Traditional learning-based approaches face two challenges. Firstly, natural language instructions often lack explicit task planning. Secondly, extensive training is required to equip models with knowledge of the task environment. Previous works based on Large Language Model (LLM) either suffer from poor performance due to the lack of task-specific knowledge or rely on ground truth as few-shot samples. To address the above limitations, we propose a novel approach called Progressive Retrieval Augmented Generation (P-RAG), which not only effectively leverages the powerful language processing capabilities of LLMs but also progressively accumulates task-specific knowledge without ground-truth. Compared to the conventional RAG methods, which retrieve relevant information from the database in a one-shot manner to assist generation, P-RAG introduces an iterative approach to progressively update the database. In each iteration, P-RAG retrieves the latest database and obtains historical information from the previous interaction as experiential references for the current interaction. Moreover, we also introduce a more granular retrieval scheme that not only retrieves similar tasks but also incorporates retrieval of similar situations to provide more valuable reference experiences. Extensive experiments reveal that P-RAG achieves competitive results without utilizing ground truth and can even further improve performance through self-iterations. We will release the source code to the public.

## CCS CONCEPTS

• **Computing methodologies** → *Robotic planning*.

## KEYWORDS

Embodied AI, Large Language Model, Progressive Method, Retrieval Augmented Generation

**ACM Reference Format:**
Anonymous Author(s). 2024. P-RAG: Progressive Retrieval Augmented Generation for Planning on Embodied Everyday Tasks. In *Woodstock '18: ACM Symposium on Neural Gaze Detection, June 03–05, 2018, Woodstock, NY Proceedings of the 32nd ACM International Conference on Multimedia (MM'24), October 28-November 1, 2024, Melbourne, Australia.* ACM, New York, NY, USA, 9 pages. https://doi.org/10.1145/nnnnnnn.nnnnnnn

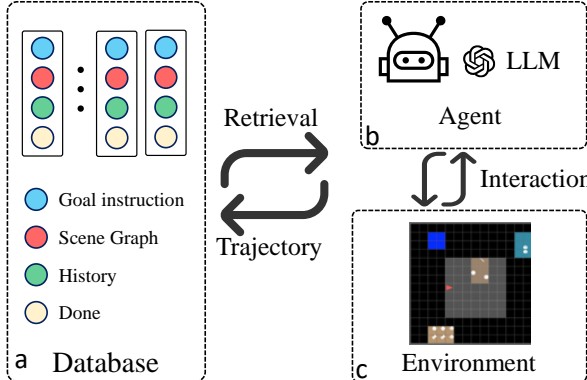

**Figure 1: The framework of our Progressive Retrieval Augmented Generation method. a) The database consists of a list of tuples, each including goal instruction, scene graph, trajectory history, and whether the task is completed. b) The agent with LLM. c) The interactive environment (MINI-BEHAVIOR or ALFRED). The database will update after each complete interaction between the agent and the environment, equipping the agent with increasingly high-quality experiences.**

## 1 INTRODUCTION

Recent years have witnessed the rapid development of Embodied AI (EAI) which aims to endow AI agents with the ability to interact with the physical world. Two famous robots in this field from Figure AI [1] and Voxposer [11] are both concentrated on Embodied Everyday Task. Due to the risks and instability inherent in conducting experiments in real environments, researchers often opt to train and test algorithms in simulation platforms. Many simulators abstract and integrate key components of the embodied everyday task into their environments, such as MINI-BEHAVIOR [13] and ALFRED [23]. In these platforms, embodied everyday task is set by a natural language goal instruction guiding the agent's objective or offering step-by-step guidance for aligning the robot's actions with linguistic commands.

Embodied everyday tasks often encounter three main challenges. 1) Dense reward feedback is typically absent, with the environment only signaling task completion after it has been entirely achieved, usually denoted by a reward of either 1 or 0. Tasks like "Cleaning up the kitchen only" in MINI-BEHAVIOR not only feature ambiguous goal instructions without clear subtasks, but also present a challenge for agents due to the difficulty in processing the natural language provided. 2) Another challenge for agents is the variable space of the action in the environment, where actions may not be fixed or there may be invalid actions that cannot be executed in current environment. For example, agents may use cooking or heating to process certain foods, but for everyday household items

[1] https://www.figure.ai/

such as pot plants and shoes, these actions are considered illegal. 3) Certain limitations imposed by real-world circumstances can easily be overlooked. For example, in specific environments, tables may be smaller than usual, unable to accommodate an excessive number of items (in the MINI-BEHAVIOR simulation, each cell is restricted to contain a maximum of three items). However, this information is not commonly known, and even language models trained on textual data may not be aware of such constraints.

Traditional learning-based approaches such as reinforcement learning (RL) can enhance the ability of the model for specific tasks and environments through iterative processes. But they often lack the capability to understand language instructions. The application of large language models (LLM) to embodied everyday task is a highly promising direction. Previous works [9, 29] have mostly focused on prompt design, which incorporates general commonsense knowledge and addresses the issue of understanding language instructions. Nevertheless, they still lack the ability to possess knowledge specific to particular tasks and environments.

To address the issue of understanding linguistic instructions and imbuing knowledge about specific tasks and situations, we propose a new framework called **Progressive Retrieval Augmented Generation** (P-RAG) for embodied everyday tasks. It is based on LLM, and designs progressive retrieval to assist in generating actions with specific contextual knowledge iteratively, as shown in Fig. 1. The progressive mechanism can iteratively increase the success rate similar to learning-based approach, without involving any training steps. With the text understanding ability of LLM, P-RAG combines both the advantages of both learning-based approaches and pre-trained LLM for planning approaches. Indeed, previous works like LLM-planner [28] have also employed retrieval augmented generation with ground few-shot to enhance the agent's knowledge of specific environments and situations. In comparison, P-RAG has improvements in the following aspects: 1) Instead of using ground-truth action list as few-shot samples, P-RAG utilizes data generated through straightforward interactions with the environment, which is more general for real scenes. 2) In the query, we not only consider searching for trajectory information related to similar tasks but also take into account trajectory information corresponding to similar situations, which provides more valuable context for LLM.

We validate the effectiveness of P-RAG in planning with extensive experiments for everyday tasks. P-RAG outperforms existing methods in few-shot setting, and it provides an effective approach that can further enhance the online planning performance for embodied everyday tasks. Additionally, P-RAG showcases its ability to generalize across different tasks, enabling it to effectively operate in various planning task. Our contributions can be summarized into the following three points. 1) We introduce a new framework for planning with LLM in embodied everyday task. This framework combines the advantages of LLM's prior knowledge and language processing capabilities enhancing the efficiency of utilizing inter-action data. 2) Instead of relying on ground truth actions as previous methods do, P-RAG enhances its performance solely through historical trajectories obtained from interaction of last round. 3) P-RAG outperforms existing methods in utilizing few-shot training datasets, and even provides a self-iteration approach that further enhances performance of testing tasks.

## 2 RELATED WORK

### 2.1 Embodied Everyday Task

Embodied Everyday Task aims to establish human-centered AI [16] that "serves human needs, goals, and values" by simulating tasks from human daily lives, including navigation and manipulating. Due to the increasing demand for human-computer interaction, tasks oriented towards ordinary users are becoming more popular, giving rise to some new tasks such as language-conditional navigation and manipulation [4, 8, 16, 26, 40, 41]. These tasks typically involve describing a desired final state using language, requiring algorithms to plan and decompose tasks. The execution process also involves a combination of navigation and manipulation operations. Recently, mainstream approaches to manipulation tasks can roughly be divided into three categories as following: 1) learning-based approaches which needs to train agents in simulation environments [19, 33, 37]. 2) LLM-based approaches which leverage the vast knowledge embedded within large language models [9, 14, 17]. 3) hybrid methods [22] combined with the above two approaches where LLMs facilitate task decomposition into subtasks, while meta-skills are honed through training to optimize subtask execution strategies.

### 2.2 Retrieval Augmented Generation

Retrieval-augmented generation is an effective tool for reducing the hallucination of large language models like GPT-4 [1] and improving their performance in generating authenticity. Initially, RAG translates the task description into a query for database retrieval. Once the query results are obtained, it integrates them with a prompt for the language model, initiating the language model to generate the final output. Initially introduced by [15], this primitive direct approach later came to be known as Naive RAG (Retrieval-Augmented Generation). It is widely used in various NLP tasks, such as narratives generation [34], abstract generation [5, 6, 32] and code generation [35, 38].

Retrieval Augmented Generation for planning represents a promising direction. However, only a limited amount of work has been conducted in this area. Previous methods like LLM-Planner [28] and RAP [36] have utilized the Retrieval Augmented Generation (RAG) approach for planning. However, in their methodologies, ground-truth planning instruction is employed during the training phase or is either utilized during the few-shot learning phase. There has not yet devised a unified approach for interactive tasks without ground truth guidance, which is a more common setting of interactive task like MINI-BEHAVIOR [13].

### 2.3 LLM for Planning

Large language models such as GPT-3.5 and GPT-4 [1], have significantly advanced the development of various applications. Compared to common text-only tasks, robotic tasks require more prior knowledge and understanding, crucially involving interaction with the physical world [10, 12, 18, 25]. Large Language Models (LLMs) are commonly employed for high-level agent planning [29]. LLMs applied in planning exhibit distinct two advantages [30] as following. Firstly, pre-trained on textual data, LLMs inherently incorporate substantial prior knowledge concerning the physical world,

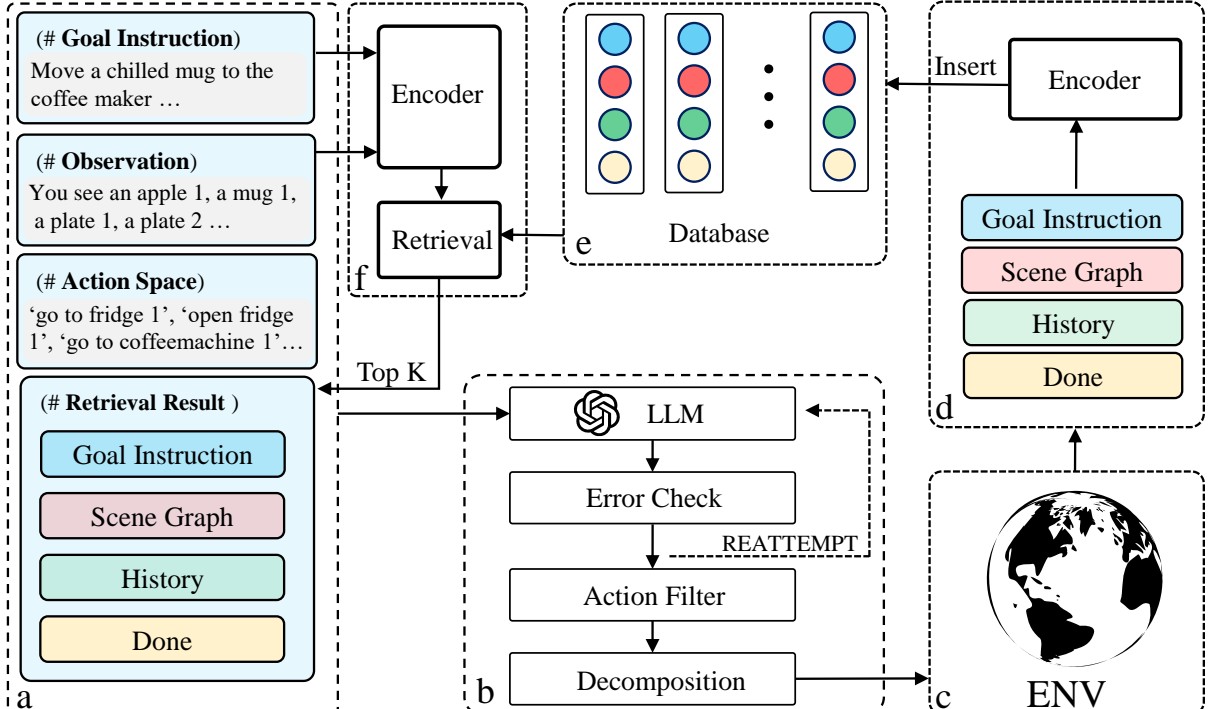

**Figure 2: The pipeline of P-RAG in each iteration. "#" stands for the form of text. a) The information transmitted to the agent consists of the following four parts: natural language goal instruction, observations obtained from the environment, the action space of the agent, and the retrieval results from the dataset. b) The agent adopts an LLM to plan a series of actions according to the information in (a). If the LLM produces unsatisfactory content, the agent will initiate a reattempt; otherwise, it will utilize a filtering mechanism to extract the requisite actions from the fields. c) The environment receives actions from the agent and returns observations, along with a "done" state denoting whether the task is completed. d) Following the completion of each iteration comprising multiple tasks, the database undergoes an update procedure. During each update, it stores the embedding vector of the goal instruction and the scene graph obtained through observation. e) The database contains the trajectories of previous iterations. f) The interface between the database and the agent's information involves two main components. Firstly, the current goal instruction and observation of agent are embedded into vectors, which are further used as query in retrieval augmented process. Secondly, the similarity between query and each database item is computed, and the top K relevant database items are returned to agent.**

enabling them to have a fundamental understanding of common robotic tasks. Secondly, LLMs excel in handling tasks involving language instructions, which are prevalent in current popular applications.

However, LLMs for planning also encounter several challenges: 1) LLMs possess only general knowledge and lack task-specific information concerning specific environments and tasks. 2) the robustness of LLMs' action outputs is inadequate, sometimes failing to adhere precisely to the required format. 3) LLMs lack comprehensive modeling of the physical world, resulting in decisions that may contravene physical constraints.

## 3  METHOD

### 3.1  Overview

To accomplish embodied everyday task, we propose a new progressive retrieval augmented planning framework named P-RAG, which integrates better task-specific knowledge progressively into the prompt of large language models (LLMs) for planning. The framework of P-RAG is shown in Fig. 1. During each interaction

episode, the trajectory of agent is collected to iteratively update a dynamic database, which is retrieved to provide task-relevant knowledge from the previous completed interaction.

Specifically, the detailed pipeline in each episode is shown in Fig. 2. First, the agent is provided with four types of information including natural language instruction, observations, the action space, and the retrieval results from database. Then, this information is input to LLM for planning, which outputs a series of actions for solving the embodied every task. After that, the environment responds to the receiving actions and converts to the new state, which provides agent new observations and reward. These new trajectory information for agent will be used to update the database, which stores the timely agent experience. In the next interaction episode, these experience knowledge in database could be retrieved according to the agent information, which returns the task-relevant knowledge to LLM for better planning. This pipeline could be progressively iterated, and increasingly improve the planning ability by providing more success trajectory information. The detailed

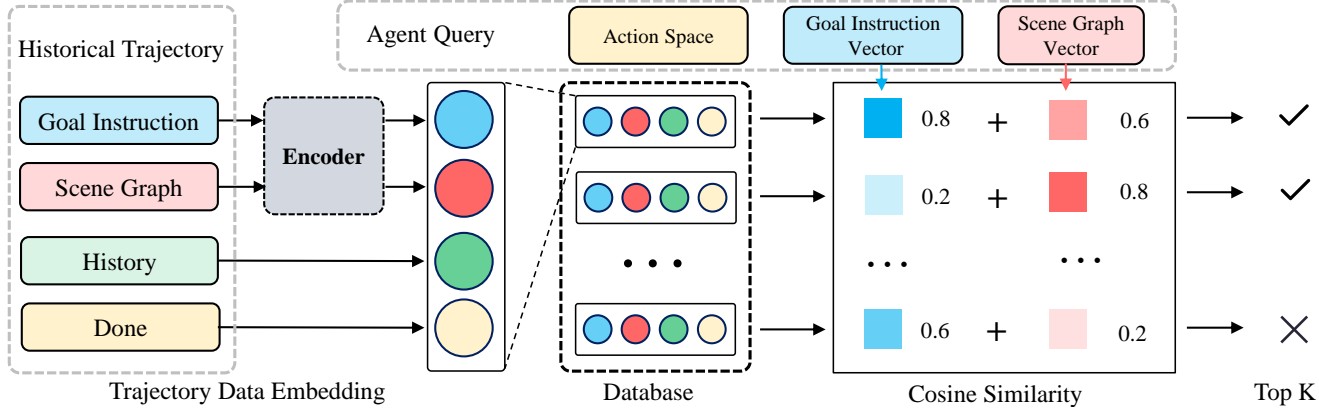

Figure 3: Database Construction and Retrieval. In P-RAG, both the construction of database and retrieval utilize encoding. 1) During the insertion process, four components are inputted: goal instruction, scene graph, history, and done. Among these, goal instruction and scene graph need to undergo sentence embedding to be stored as vectors in the database. 2) When retrieval is required, the current task's goal instruction and scene graph are used as queries. They are also encoded to sentence embedding, while simultaneously computing the similarity score between them and the corresponding vectors in the database. The top K historical trajectory information is then returned based on the aggregated similarity scores.

components of this pipeline will be elaborated in the following subsections.

## 3.2 Agent Input Information

Four parts of information will be provided to the agent: goal instructions, observations, action space, and retrieval results. Each part of the information will ultimately be transformed into text format and then integrated into coherent prompt delivered to agent.

Initially, during the interaction with the task environment, the current task's goal instruction will be provided, such as "Move a chilled mug to the coffee maker". During the interaction between P-RAG and the environment, before the agent provides an action, it will first return an observation image to depict the current observed state. We convert the observation image into a scene graph format, which is easier for LLMs to process. We extract the instance labels of all objects from the image and detect the relative relationships between each object, such as "on the top of", "inside of", and so on. In detail, we slightly adjust the extraction of the scene graph for different interaction environments. For ALFRED, we utilize tools from ALFWORLD, which include interactive TextWorld environments [7] that mirror embodied worlds in the ALFRED dataset [23], to extract instance-level labels of objects observed in the image directly in front of the robot in the environment. After obtaining the instance labels, we first identify the landmarks among them and designate them as key nodes in the scene graph. Next, we organize these labels into a list format, with the key node being placed as the first node in the list for subsequent encoding steps. For the MINI-BEHAVIOR environment, which utilizes gym-minigrid, we sequentially compare the relative relationships between each object by function API in the environment to obtain a relationship matrix with form of each cell in matrix like "object_1/object_2/relationship: True/False". After obtaining the relationship matrix, to reduce the length of the context input to the LLM, P-RAG retains only those pairs that are true. The action space is also a crucial component,

and the action space provided to the LLM consists of high-level actions. Subsequent steps in section 3.3 will involve instruction decomposition, where high-level instructions are converted into low-level actions to be provided to the environment.

Another component involves historical trajectory information retrieved from database to provide the agent with references, thereby increasing its task-specific knowledge. The retrieval result includes goal instructions, scene graphs, history, and the done state from the previous interaction history. Among them, the goal instruction and scene graph are in the same format as mentioned above. "History" refers to the sequence of $(A_t, O_t)$ pairs, providing the agent with entire pairs in the task of previous round. The "done" states indicate whether the task corresponding to these historical pieces of information was completed by the agent at that time.

## 3.3 Planning with LLM

The four components of information mentioned above in section 3.2 will be integrated into a single coherent prompt and provided as input to the LLM. Fig. 2 part b illustrates the main workflow of this section, including the LLM, error check, action filter, and decomposition components. For the first component, we choose GPT-4 and GPT-3.5, renowned for their widespread utilization across various generation tasks, to serve as LLM in P-RAG. The second part involves error checking, where the input consists of the text output from the LLM. It employs regular expression matching and compares the generated actions with those in the action space to ensure adherence to fixed formatting and validity. In the case of invalid actions or non-compliant formatting, a new requirement will be raised for the LLM. Otherwise, the output from the LLM will be passed on to the next component. The third component is the action filter, which extracts the necessary strings from the text generated by the LLM using regular expression matching, providing high-level actions. The final component is decomposition, which translates the high-level actions obtained from the previous component into

low-level actions to be executed in the environment. For example, the "navigation" high-level action is decomposed using the Fast Marching Method (FMM) [21].

Specifically, FMM is a numerical technique designed to tackle boundary value problems arising from the Eikonal equation [21]. We employ a simplified and customized FMM as the tool for decomposing the navigation component from high-level action to low-level action in our process. The FMM process used in P-RAG can be summarized as follows: Firstly, we construct a flat model of the environment with different navigable positions. In MINI-BEHAVIOR, we determine whether each cell in the observation can be covered by the agent. Secondly, we set the current point as the source point and the destination point as the sink point. Then, we compute the Euclidean distance from each point to the source point. Finally, starting from the sink point, we search for the point with the minimum distance value within its one-step neighborhood to serve as the next subgoal point.

## 3.4 Database Construction and Retrieval

P-RAG interacts with the environment through agents to construct and update a historical trajectory database. Then, it retrieves information from the database to provide as reference for the agent. It iterates over a set of tasks across multiple rounds. The first round is dedicated to constructing the database. In each subsequent round, P-RAG utilizes retrieval to enhance the capability of action generation and leverages the interaction history of each round to update the database.

The database is initialized as an empty collection. The agent interacts with the environment exclusively through goal instructions and observation obtained from each step. After the completion of all task interactions in the first iteration, a round of interaction between the agent and the environment yields valuable information. This information increases the agent's knowledge of specific tasks. The database will store this information and update itself with the latest historical trajectory data after each subsequent iteration.

During each round of data update, the database will retain four key pieces of information for each task interaction sequence: the goal instruction, scene graph, historical data, and completion status. The details of database construction and retrieval pipeline is illustrated in Fig. 3. When an agent interacts with the environment during a task, it first receives the environment's goal instruction $I_g$ and observation $O_t$. Then it encodes with MiniLM [31] both of them with formula as

$$Q_{goal} = Encode(I_g), \tag{1}$$

$$Q_{obs,t} = Encode(\mathcal{F}(O_t)), \tag{2}$$

where $Q_{goal}$ and $Q_{obs,t}$ represent the query embedding vector encoded from goal instruction $I_g$ and observation $O_t$ at time t. $\mathcal{F}$ means scene graph extraction module. With these embedding representation of goal instruction and observation, we use cosine similarity to retrieval the most similar task trajectory in task name or in situation. Each set of embedding vectors serves as keys in the historical records, and will undergo similarity computation with the query. The similarity score for the $n$-th interaction in a task dialogue $s_t$ is calculated as following:

$$s_n = sim(Q_{goal}, K_{goal}) + \max_{t \in [1,N]} sim(Q_{obs,n}, K_{obs,t}), \tag{3}$$

---

**Algorithm 1** P-RAG for Planning

---
**INPUT:** $ENV_i, i = 1, 2, ..., N_{env}$
**OUTPUT:** $A_{i,t}^{i=1,2,...,N_{env}}_{t=1,2,...,N}$
  $DB \leftarrow \varnothing$;
  $ENV_i \leftarrow RESET, i = 1, 2, ..., N_{env}$;
  **for** $iter in [1, ITER]$ **do**
    **for** t in [1,N] **do**
      $O_{i,t}, I_{i,goal} \leftarrow ENV_i(A_{i,t}), i = 0, 1, 2, ...N_{env}$;
      $\leftarrow ENV_i, i = 0, 1, 2, ...N_{env}$;
      $Q_{obs,i,t} \leftarrow Encode(\mathcal{F}(O_{i,t}))$;
      $Q_{ins,i} \leftarrow Encode(I_{i,goal})$;
      **if** $DB \neq \varnothing$ **then**
        $K_{obs,i,t}, K_{ins,i} \leftarrow DB, i = 1, 2, ...N_{nev}$;
        $s_i \leftarrow equation\ 3$;
        $I_{goal}, SG, H, D \leftarrow TOPK(DB, \{s_i\}_{i=1}^{N_{env}})$;
        $A_{i,t} \leftarrow Agent(LLM, I_{goal}, SG, H, D)$;
      **else**
        $A_{i,t} \leftarrow Agent(LLM)$;
      **end if**
    **end for**
  **end for**

---

where $Q_{goal}$ and $K_{goal}$ denote the embedding vectors corresponding to the goal instructions, while $Q_{obs,n}$ and $K_{obs,t}$ represent the embedding vectors of the scene graph for the $n$-th interaction and the $t$-th interaction, respectively. We select the maximal similarity scores in the $t$-interaction for scene graphs. Ultimately, within the database, P-RAG selects the top K entries as output based on composite score that incorporates both task and scene graph similarities.

## 3.5 Progressive Iteration

As depicted in Algorithm 1, P-RAG initializes an empty database and initiates successive iterations. At this stage, P-RAG makes decisions solely based on observations of the environment and the generic prior knowledge of LLM. After the first iteration, the historical information from the previous iteration is collected and stored in the database. The database is progressively updated when the agent trajectories of new round are completed. In each subsequent iteration, P-RAG utilizes the collaborative similarity retrieval of scene graph and task name to identify similar tasks and scenes across different tasks. It then provides this information to LLM to facilitate more informed decision-making. The progressive approach is widely employed in learning-based solutions, yet its application in the RAG (Retrieval-Augmented Generation) framework for planning remains a novel paradigm. In scenarios where ground truth is not available, the progressive approach offers a mode of gradual performance improvement compared to the direct method. Moreover, it allows for the accumulation of environment-specific knowledge for the agent through historical trajectories.

## 4 EXPERIMENT

### 4.1 Experimental Setup

**Dataset**. We select two datasets, MINI-BEHAVIOR [13] and AL-FRED [23], for our experiments. Extracting 20 activities from BE-HAVIOR 1K and abstracting them into a grid environment, the MINI-BEHAVIOR platform offers a comprehensive array of tasks

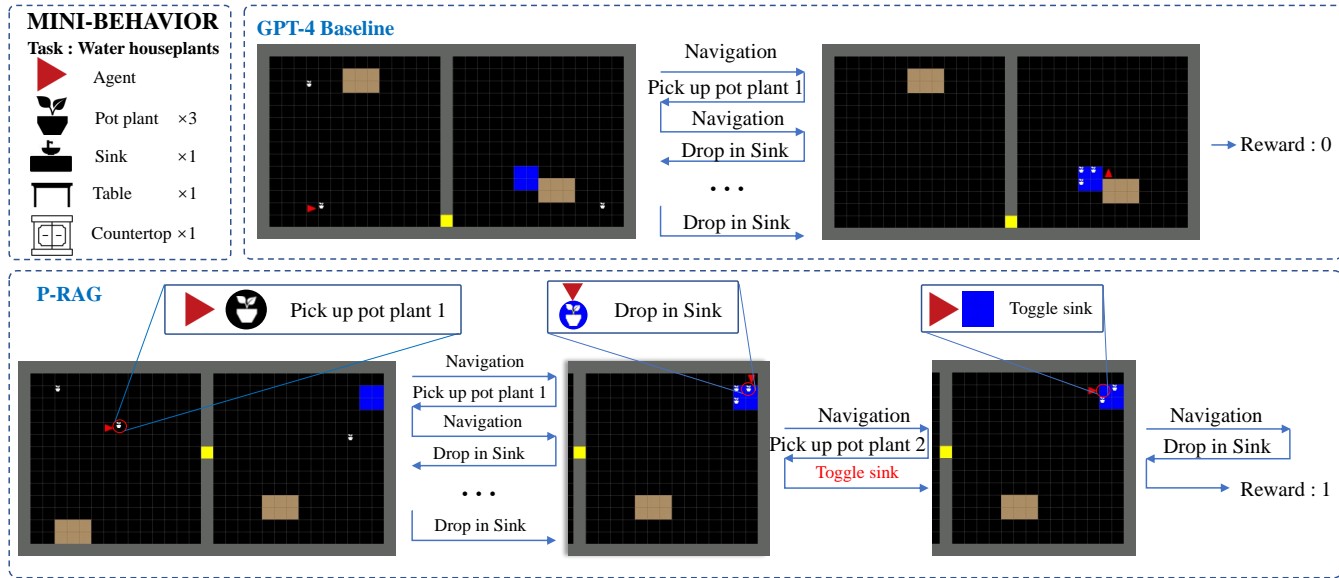

**Figure 4: Comparison on planning trajectories between GPT-4 baseline and P-RAG. The baseline method follows a decision process of sequentially picking up three pot plants and placing them in the sink, considering the task complete. However, it fails to achieve the task successfully. In contrast, P-RAG utilizes comprehensive historical trajectory information to make decisions, leading to the judgment to toggle the sink and ultimately accomplishing the task.**

for agents to navigate and manipulate over extended periods. The agent needs to input linguistic instructions and then make decisions to output actions, such as turning left/right, picking up, and so on. Compared to MINI-BEHAVIOR, ALFRED offers more tasks for evaluating the agent and provides realistic observation images. In the ALFRED dataset, we employ the text-world tool from ALFWORLD [24] to convert image-based observations into text format. Since the MINI-BEHAVIOR dataset only consists of 20 environments, we conduct evaluating all of them directly in the experiment. The ALFRED dataset is divided into Valid Seen, Valid Unseen and Train datasets. Following LLM-Planner [28], we choose 100 task environments from a pool of 21,023 examples in the ALFRED training set for interaction by P-RAG, which is called Train100 in our experiments. Ensuring a balanced representation, we employ random stratified sampling across all seven task types following the setting to the approach adopted by the LLM-Planner. Valid Seen and Valid Unseen consist of 242 and 85 tasks [23], respectively.

**Implementation Details.** Different from the previous works [28], we do not require a ground-truth training dataset. All we need is interaction with the environment as the evaluation stage. P-RAG is designed to provide a unified framework for conducting experiments across different datasets. We use the GPT series model as our LLM planner, with the majority of tests utilizing GPT-4 [1]. In the P-RAG setup, the database will store historical trajectories from previous rounds, with a maximum of 6 iterations. For the ALFRED dataset, we use the JSON format to organize the embedding vector with 384 dimensions of goal instruction and scene graph for each task. The raw trajectory information is recorded in text format in a SQLite[1] database, with an average size of 3.2K bytes per task.

---

[1]https://www.sqlite.org/

## 4.2 Comparison Experiment with the State-of-the-art Methods

We first conduct evaluations on the ALFRED dataset to compare the success rates (SR) of the state-of-the-art methods [2, 3, 20, 27, 28, 39] and our method. To ensure fair comparison, after 3 iterations in the training dataset, P-RAG evaluates its performance on both test datasets. In particular, we curate a subset of 100 task environments from the training set for agent to interact. The database undergoes three iterative updates prior to subsequent evaluation on the two evolution datasets.

As illustrated in Table 1, we categorize different methods into two groups based on experimental settings. One group including HiTUT [39], HLSM [3], E.T. [20] and M-TRACK [27] utilizes the full train set with 21,023 instruction and trajectory pairs to derive performance results. Another group including HLSM [3], LLM-Planer [28] and Saycan [2] utilizes a smaller subset of training data. Note that within the experimental setup, certain methods denoted with a star (*) utilize step-by-step instructions rather than goal instructions, such as Saycan [2] and E.T. [20]. The performance comparison in the table reveals that our approach denoted as "P-RAG (Ours)" outperforms existing state-of-the-art methods of utilizing few training dataset on both Valid Seen and Valid Unseen dataset. These results demonstrate the effectiveness of our method, which develops the progressive retrieval augmentation to assist the large language model to obtain task-relevant information. Compared to methods trained on the full dataset, P-RAG maintains high performance even under conditions where only approximately 1/200 of the training data is utilized. These results prove that our method possesses better generalizability than these training methods, which usually overfit the training datasets and achieve better performance on the Val Seen dataset than Val Unseen dataset.

| Model | Dataset | G.T. | Valid Unseen | Valid Seen |
|-------|---------|------|--------------|------------|
| HiTUT [39] | Full | ✓ | 10.23 | 18.41 |
| HLSM [3] | Full | ✓ | **18.28** | 29.63 |
| *E.T. [20] | Full | ✓ | 7.32 | **46.59** |
| *HiTUT [39] | Full | ✓ | 12.44 | 25.24 |
| *M-TRACK [27] | Full | ✓ | 17.29 | 26.70 |
| *HLSM [3] | Part | ✓ | 0.00 | 0.13 |
| LLM-Planer [28] | Part | ✓ | 12.92 | 13.53 |
| *Saycan [2] | Part | ✓ | 9.88 | 12.3 |
| GPT-4 [1] | - | ✗ | 7.05 | 17.46 |
| **P-RAG (Ours)** | Part | ✗ | **14.11** | **18.2** |
| **P-RAG (Self-Iter.)** | - | ✗ | **27.4** | 19.05 |

Table 1: Comparison of Different Method on ALFRED. Star (*) stands for using step-by-step instruction instead of goal instruction. G.T. means ground-truth action. In the "Dataset" column, "Full" indicates the utilization of the entire training dataset, while "Part" indicates sampling from a subset of the training dataset. "P-RAG (Self-Iter.)" represents the iterative updates of P-RAG on the same group of task datasets.

Besides the standard setting of our method, we also construct a self-iteration variant denoted as "P-RAG (Self-Iter.)". It constructs progressive iteration over the test dataset, which means the retrieval database is constructed by the trajectories on test dataset. It directly improves the success rates of testing tasks, since the agent obtains better task-relevant experiences about testing tasks by progressive retrieval. From Table 1, with self-iteration on the Valid Unseen dataset, our method denoted as "P-RAG (Self-Iter.)" can even outperform all the methods, both with and without using the entire training dataset, which further verifies the effectiveness of the proposed progressive retrieval augmented planning framework.

We also evaluate the performance of P-RAG on MINI-BEHAVIOR which lacks ground-truth actions annotation, and therefore can be used to evaluate our method and compared methods in weak supervision setting. As shown in Table 2, we evaluate through three evaluation metrics: Total success rate (SR), which represents the success average by episodes; Task success rate (SR), indicating success average by tasks; Success weight by Path Length (SPL), evaluated according to the following formula:

$$SPL = \frac{1}{N} \sum_{i=1}^{N} S_i \frac{L_i}{max(L_i, P_i)}, \tag{4}$$

where N is total number of evalutional episodes, $S_i$ represents whether the current episode is successful, $L_i$ denotes the number of actions used by the agent, and $P_i$ represents the number of steps needed to complete the task in the shortest possible manner.

We compare the success rate of P-RAG relative to its corresponding baseline with LLM selecting either GPT-4 or GPT-3.5. From the results in the table, we can observe that P-RAG demonstrates significant improvements of 1.7% and 2.5% compared to the baselines of GPT-4 and GPT-3.5, respectively. Although simple and lightweight, MINI-BEHAVIOR presents a formidable challenge for popular Reinforcement Learning algorithms, particularly in the absence of a dense reward signal. The vanilla PPO algorithm is only able to achieve a valid success rate (approximately 8%) after training 1e6 steps and evaluating on a single task (not work in other tasks) within MINI-BEHAVIOR [13]. In contrast, P-RAG

| MINI-BEHAVIOR | GPT-4[1] | P-RAG-4 | GPT-3.5 | P-RAG-3.5 |
|---------------|----------|---------|---------|-----------|
| Total SR | 15% | 16.7% | 7.5% | 10% |
| Task SR | 20% | 25% | 20% | 20% |
| SPL | 13.8% | 15% | 7.5% | 9.5% |

Table 2: Comparison of Retrieval Augmented Models on MINI-BEHAVIOR. GPT-3.5 and GPT-4 represent the results of each as the baseline LLM planner. P-RAG-3.5 and P-RAG-4 represent the results of setting the LLM in P-RAG as GPT-3.5 and GPT-4, respectively.

| Dataset | Original | 1st Iter. | 2nd Iter. | 3rd Iter. |
|---------|----------|-----------|-----------|-----------|
| Train100 | 5% | 9% | 10% | 11% |
| Valid Unseen | 7.05% | 14.11% | 20.00% | 22.35% |

Table 3: Success Rate with the Iteration Number on ALFRED.

achieves 16.7% Total SR requiring iterations of no more than 6 eposides and accomplish 5 tasks (compare to single task of vanilla PPO algorithm), demonstrating its effectiveness across different environments and its few-shot property. We select one group of task trajectories as visualization result, as shown in Fig. 4. The visualization of trajectory demonstrates the decision-making processes of P-RAG and the GPT-4 baseline for the task "water houseplants". From the visualization results, it can be observed that P-RAG is able to make judgments with more task-specific knowledge by referring to historical trajectory information.

## 4.3 Ablation Study & Parameter Analysis

**Progressive Iteration**. We conduct additional experiments on P-RAG regarding self-revolution to validate its progressive effect. Compared to the previous approach of iterating on the training dataset, we directly conduct progressive retrieval iteration on the same dataset, which we call self-iteration. In particular, we choose to conduct self-iteration on the Valid Seen dataset and training dataset (where there may be overlap between the Valid Seen and Train datasets), respectively. From Table 3, there is an improvement after progressive iteration in both task datasets. P-RAG not only demonstrates the advantages of performance in standard experimental setting as shown in the first row "Train100", but also achieves performance improvement through progressive iteration on the Valid Unseen dataset as shown in the second row in Table 3.

We also conduct in-depth experiments on performance saturation, by enhancing retrieval through updating historical information from the previous round after encoding it into the database. Subsequently, we update the database based on the current round's historical trajectory, iterating multiple times until saturation is achieved, as shown in Fig. 5. From Fig. 5, P-RAG achieves significant performance improvement through iteration. In the ALFRED Valid Unseen dataset, the success rate is improved from 7.05% at the beginning to 27.4% after 5 rounds of iteration. Similarly, in the ALFRED Train 100 dataset, the success rate increases from 5% at the beginning to 11% after 3 rounds of iteration. Both of them all eventually reach performance saturation. From the curves, it can be observed that the curve of success rate increase in each iteration of P-RAG until convergence, indicating that it is approaching the performance limit of LLM as a Planner on the testing tasks.

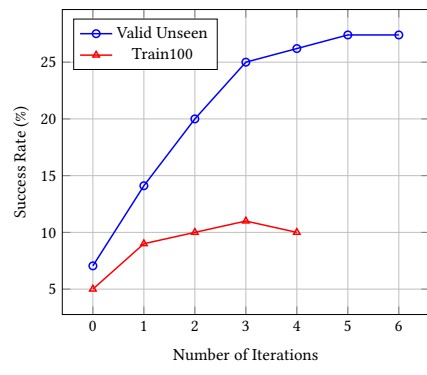

Figure 5: Performance with Iteration Number on ALFRED.

**Correlation Between Iteration**. We also continue to study the correlation between the successfully completed tasks in the iterative process of P-RAG. Table 4 displays the correlations between tasks completed in different iteration stages. The data values in the table are derived from the results with iterations number of 0, 1, 2, 3 in the ALFRED dataset. Each pair of True and False corresponds to successful and failed tasks in a single iteration. The value within each cell indicates the ratio of occurrences of "done" compared to the total occurrences of True/False in the corresponding row's iteration. Considering the value located in the top-left corner of the table, this value signifies the proportion of successful tasks from the initial iteration that remain successful after the first subsequent iteration.

| | | Next 1 Iter. | | Next 2 Iter. | | Next 3 Iter. | |
| | | True | False | True | False | True | False |
| --- | --- | --- | --- | --- | --- | --- | --- |
| **0th Iter.** | **True** | 83.3% | 16.7% | 100% | 0.0% | 83.3% | 16.7% |
| | **False** | 6.8% | 93.2% | 14.9% | 85.1% | 17.6% | 82.4% |
| **1st Iter.** | **True** | 72.7% | 27.3% | 72.7% | 27.3% | - | - |
| | **False** | 12.3% | 87.7% | 13.7% | 86.3% | - | - |
| **2nd Iter.** | **True** | 88.2% | 11.8% | - | - | - | - |
| | **False** | 4.5% | 95.5% | - | - | - | - |

Table 4: Relation Between Iteration on ALFRED. "N-th Iter." represents the completion status of tasks after the n-th iteration of P-RAG. For the row labeled "Next m-th." each cell represents the comparison between the (n+m)-th iteration and the n-th iteration. The value in each cell represents the proportion of True/False occurrences of "done" in its column compared to the total occurrences of True/False in its corresponding row's iterations.

By analyzing the data in Table 4, it is evident that P-RAG predominantly achieves success in subsequent iterations based on the success of the previous iteration. Furthermore, it also demonstrates the capability to achieve success in tasks that were unsuccessful in the previous iteration. This highlights how P-RAG not only effectively maintains the performance from the previous iteration but also leverages information from unsuccessful trajectories of the previous iteration to achieve new successes. This meticulous demonstration underscores the sources of performance enhancement in P-RAG.

| P-RAG (Self-Iter.) | | | P-RAG | P-RAG w/o SG | GPT-4 [1] |
| --- | --- | --- | --- | --- | --- |
| 1st | 2nd | 3rd | | | |
| 14.11% | 20.00% | 22.35% | 14.11% | 13.10% | 7.05% |

Table 5: Ablation with Scene Graph on ALFRED Valid Unseen. The first three columns represent P-RAG with 3 rounds of self-iterations on Valid Unseen dataset. The following two columns indicate testing on Valid Unseen after iterations on Train100.

**Ablation Study**. We demonstrate through ablation experiments to verify 1) the effectiveness of the progressive approach and 2) the effectiveness of the joint retrieval of scene graph and task name compared to using task name alone.

For the former, in the last row of Table 1, P-RAG is demonstrated to be effective through iterations on both Valid Unseen and Valid Seen. Through iterations, P-RAG outperforms nearly all the methods, including step-by-step instruction and training on the full dataset, in performance on Valid Unseen without training and with a few number of samples. Similarly, in Table 2, conducting two sets of experiments with LLM as GPT-4 and GPT-3.5 also demonstrates the effectiveness of the progressive method.

To further evaluate the effectiveness of the progressive mechanism, we conduct extra ablation experiments on the ALFRED Valid Unseen dataset, as shown in Table 5. The interpretation of the data in the table is as following. The "GPT-4" column represents the results by using the GPT-4 baseline for testing. The "P-RAG (Self-Iter.)" column represents the results obtained through testing with 1, 2 and 3 iterations, respectively. The "P-RAG" column represents the results obtained by iteratively using task name and scene graph joint retrieval on the Train100 dataset and then testing on the Valid Unseen dataset. The "P-RAG w/o SG" column represents the results obtained by solely using task name for retrieval, iteratively on the Train100 and then testing on the Valid Unseen dataset. From the table, it is evident that compared to not utilizing scene graph for joint retrieval, P-RAG exhibits significant improvements. This demonstrates the effectiveness of the progressive mechanism and the necessity of incorporating scene graph and task name retrieval.

## 5 CONCLUSION

We propose a novel progressive retrieval augmented generation framework called P-RAG for planning to address the issue of lacking task-specific and scene-specific knowledge in LLM. P-RAG distinguishes previous methods by eliminating the need for ground truth and achieves promising performance with just a few task interactions. By collaboration with task name and scene graph for P-RAG to retrieval, we ensure historical records selected from database are not only similar in tasks but also in situations. P-RAG's unique iterative capability continuously acquires historical information through interaction. By leveraging enhanced retrieval capabilities, P-RAG incrementally accumulates task-specific knowledge during iterations, thereby improving its planning performance. The experiments demonstrate the effectiveness of P-RAG compared with the state-of-the-art planning methods for Embodied Everyday task. We hope that the ground-truth-free capability in P-RAG can be applied to a broader range of planning tasks and deliver even more outstanding performance.

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
