# OpenReview forum: "P-RAG: Progressive Retrieval Augmented Generation For Planning on Embodied Everyday Task"
_acmmm.org/ACMMM/2024/Conference — MM2024 Poster_

### Official Review · Reviewer_w7JW · 2024-05-17

**Rating:** 2
**Confidence:** 4

**Summary:**

The paper introduces a novel approach named Progressive Retrieval Augmented Generation (P-RAG) designed to enhance the performance of Large Language Models (LLMs) in the context of embodied AI tasks. P-RAG addresses the limitations of traditional retrieval augmented generation methods by implementing a progressive, iterative database update mechanism. This approach allows the model to accumulate and refine task-specific knowledge over time without relying on ground-truth data. The method leverages both similar tasks and situations for retrieval, providing a richer context for the generation process. The paper claims that P-RAG offers competitive results and improves through self-iterations, with plans to release the source code publicly.

**Strengths:**

1. Innovative Retrieval Strategy: P-RAG’s iterative retrieval approach represents a significant advancement over traditional one-shot retrieval methods. By updating the retrieval database continuously, the system can refine its knowledge and adapt to new information, enhancing its decision-making capabilities.

2. Granular Retrieval Scheme: The introduction of a more granular retrieval scheme that considers both similar tasks and situational contexts is a commendable enhancement. This could lead to more nuanced and contextually appropriate responses from the model.

3. Demonstrated Empirical Results: The paper reports extensive experimental results showing that P-RAG performs competitively without ground-truth data and improves further with self-iterations. This empirical validation is crucial for establishing the efficacy of the proposed approach.

**Limitations:**

1. Lack of Detailed Comparative Analysis: The paper does not sufficiently explore how different components (like Goal Instructions and Scene Graphs) individually contribute to the overall effectiveness of the method. A detailed component-wise analysis could provide deeper insights into which aspects of the model are most impactful.

2. Unclear Scene Graph Generation Process: The generation process for scene graphs and the specifics of the encoding mechanism remain unclear. Details about whether these graphs are generated by an LLM, the version of the LLM used (e.g., GPT-4), and the involvement of any visual modules are crucial for understanding the complete workflow and effectiveness of the retrieval process.

3. Questions on Methodological Innovations: The paper's methodological innovation appears to be mainly in the data handling strategy rather than a fundamental advancement in model architecture or interaction. Clarifying how this approach differs significantly from other retrieval-augmented methods in terms of flexibility, such as adaptive weighting or more dynamic retrieval mechanisms, could strengthen the novelty claim.

Minor:
1. Expand Component-Wise Comparative Studies: Include experiments or analyses that isolate the effects of different system components like Goal Instructions and Scene Graphs. This could help identify their individual contributions and optimize the system accordingly.

2. Detail the Scene Graph Generation and Encoding Process: Provide clear information on how scene graphs are generated, which model or encoder is used, and whether any multimodal information is incorporated. This transparency will help in replicating and understanding the proposed system's functionality.

3. Clarify Methodological Innovations: Enhance the discussion on how P-RAG’s approach to retrieval and data handling offers methodological innovations beyond the evident advantages of using enriched data sources. Consider discussing potential adaptive or dynamic features that could be integrated into the retrieval process.

**Suitability:**

3

---

### Official Review · Reviewer_6ru3 · 2024-05-23

**Rating:** 4
**Confidence:** 2

**Summary:**

To carry out the Embodied Everyday task, the authors propose Progressive Retrieval Augmented Generation (P-RAG), which enhances language model capabilities by iteratively updating task-specific knowledge without relying on ground truth. P-RAG's iterative retrieval of relevant information significantly improves performance compared to one-shot methods. Extensive experiments show P-RAG achieves competitive results and improves further through self-iterations.

**Strengths:**

1. This paper uses Retrieval Augmented Generation (RAG) to enhance the performance of LLMs in Embodied Everyday Tasks, aligning with the development trend of applying LLMs to downstream tasks. Hence, the novelty of this paper is commendable.
2. The pipeline presented in this paper is comprehensive, with steps such as error checking reflecting the current state of LLM applications. This can provide valuable insights for other LLM-based research.

**Limitations:**

1. It would be beneficial to include a brief introduction of comparative methods such as LLM-Planner to highlight the effectiveness of RAG for LLM-based approaches.
2. The paper claims that this method is more effective than one-shot methods. Does the one-shot sample include chain-of-thought prompting? Can you compare it with methods using ground-truth and chain-of-thought prompting?
3. What is the scale of the test datasets? How many tokens are used in the prompts? Approximately how much inference cost does the LLM incur per instance? I recommend adding an analysis of the application costs of using LLMs.
4. TYPO: line 734 "success weight by path" should be "success weighted by path".

**Suitability:**

3

---

### Official Review · Reviewer_6kAk · 2024-05-24

**Rating:** 4
**Confidence:** 4

**Summary:**

Previous works based on Large Language Model(LLM) either suffer from poor performance due to the lack of task specific knowledge or rely on ground truth as few-shot samples. To address the above limitations, They propose a novel approach called Progressive Retrieval Augmented Generation (P-RAG), which not only effectively leverages the powerful language processing capabilities of LLMs but also progressively accumulates task-specific knowledge without ground-truth.

**Strengths:**

1. They introduce a new framework for planning with LLM in embodied everyday task. This framework combines the advantages of LLM’s prior knowledge and language processing capabilities enhancing the efficiency of utilizing interaction data.

2. Instead of relying on ground truth actions as previous methods do, P-RAG enhances its performance solely through historical trajectories obtained from interaction of last round.

**Limitations:**

1. What is the structure of the encoder in Trajectory Data Embedding? How efficient is it?

2. This paper is based on the processing of Embodied Everyday Tasks using existing LLM, and I believe it focuses more on some tricks.

3. This paper is more about a combination of existing methods, and I hope to see some unique content.

**Suitability:**

3

---

### Meta-Review · Area_Chair_ZL3t · 2024-06-30

**Recommendation:** Accept (Poster)
**Confidence:** 5

**Metareview:**

The paper receives weak accept, borderline accept, and weak reject. Two reviewers acknowledge the contribution of the paper, while another reviewer doubts the methodological innovations. After checking the paper, reviews, and rebuttal, the area chair decides to recommend accept, because the proposed progressive RAG for planning is novel and inspiring. The area chair would suggest the authors to take the reviewer comments into consideration when preparing for the camera-ready version.